# Bidirectional Control between Cholesterol Shuttle and Purine Signal at the Central Nervous System

**DOI:** 10.3390/ijms23158683

**Published:** 2022-08-04

**Authors:** Daniela Passarella, Maurizio Ronci, Valentina Di Liberto, Mariachiara Zuccarini, Giuseppa Mudò, Carola Porcile, Monica Frinchi, Patrizia Di Iorio, Henning Ulrich, Claudio Russo

**Affiliations:** 1Department of Medicine and Health Sciences “V. Tiberio”, University of Molise, 86100 Campobasso, Italy; 2Department of Pharmacy, University of Chieti-Pescara, 66100 Chieti, Italy; 3Department of Experimental Biomedicine and Clinical Neurosciences, University of Palermo, 90133 Palermo, Italy; 4Department of Medical Oral and Biotechnological Sciences, University of Chieti-Pescara, 66100 Chieti, Italy; 5Biochemistry, Institute of Chemistry, University of São Paulo, São Paulo 05508-060, Brazil

**Keywords:** purinergic receptors, LDL receptors, cholesterol

## Abstract

Recent studies have highlighted the mechanisms controlling the formation of cerebral cholesterol, which is synthesized in situ primarily by astrocytes, where it is loaded onto apolipoproteins and delivered to neurons and oligodendrocytes through interactions with specific lipoprotein receptors. The “cholesterol shuttle” is influenced by numerous proteins or carbohydrates, which mainly modulate the lipoprotein receptor activity, function and signaling. These molecules, provided with enzymatic/proteolytic activity leading to the formation of peptide fragments of different sizes and specific sequences, could be also responsible for machinery malfunctions, which are associated with neurological, neurodegenerative and neurodevelopmental disorders. In this context, we have pointed out that purines, ancestral molecules acting as signal molecules and neuromodulators at the central nervous system, can influence the homeostatic machinery of the cerebral cholesterol turnover and vice versa. Evidence gathered so far indicates that purine receptors, mainly the subtypes P2Y_2_, P2X_7_ and A_2A_, are involved in the pathogenesis of neurodegenerative diseases, such as Alzheimer’s and Niemann–Pick C diseases, by controlling the brain cholesterol homeostasis; in addition, alterations in cholesterol turnover can hinder the purine receptor function. Although the precise mechanisms of these interactions are currently poorly understood, the results here collected on cholesterol–purine reciprocal control could hopefully promote further research.

## 1. Introduction

Obesity and associated cardiometabolic diseases are closely related to altered lipid homeostasis at a peripheral level. Hundreds of scientific articles have recently been focused on the pathophysiology of these disorders and their harmful consequences as well as on the strategies aimed at limiting their incidence and diffusion worldwide [1,2,3]. Among the clinical studies on these aspects, many have pointed out that metabolic diseases can cause neuroinflammation and/or brain alterations, mainly in the cerebral vascularity, therefore, being related to brain alterations and pathologies [4]. Additionally, epidemiological studies have shown that intake of nutrients rich in fat affects cognitive functions and memory, especially in older people [5]. In line with this evidence, a recent article has confirmed that a Western diet with high content of lipids and fructose, administered to middle aged rats, is associated with the alteration of cholesterol levels and beta amyloid metabolism, mostly in the hippocampus of those animals [6]. 

In this scenario, it is important to highlight that the brain is made of lipids similar to the adipose tissue. Therefore, there must be also local mechanisms assuring cerebral lipid homeostasis, which is crucial for maintaining normal brain activities. The brain, indeed, is the main organ in the human body that assures life, regulating unconscious vital functions at the cardiovascular and respiratory levels, as well as voluntary nervous activities. 

Brain lipids are sphingolipids, glycerophospholipids and cholesterol, which are present in almost equal ratios. Here, we will focus on cholesterol, which represents about 25% of total body cholesterol [7,8]. It is a fundamental component of neural cell membranes and contributes with the other lipids aforementioned to form the myelin sheath, enwrapping and protecting neurons. However, cholesterol performs several other functions, as it is required for synapse and dendrite formation [9,10], and axonal guidance [11], thus, assuring normal synaptogenesis and neuronal plasticity [12]. Cholesterol is also a precursor of neuroactive steroids [13] and it is indispensable for presynaptic vesicle formation and postsynaptic receptor expression [14]. 

Since peripheral lipoproteins delivering cholesterol cannot pass the blood–brain barrier (BBB), most of this lipid is synthesized in situ, even if very few amounts may be imported from the periphery [15]. Therefore, it is clear that brain lipid homeostasis can be mainly lost as a consequence of local alterations of lipid turnover. Accordingly, defects in cholesterol turnover lead to neurodegenerative diseases including Alzheimer’s disease (AD), Parkinson’s disease (PD), Huntington’s disease (HD) and Niemann–Pick C disease (NPC) [16,17,18].

The main source of cerebral cholesterol are glial cells, in particular astrocytes, which produce it three times more than neuronal cells [19]. Astrocytes synthesize and secrete cholesterol linked to specific apolipoproteins belonging to the E family, via ATP-binding cassette (ABC) transporters. Adult neurons, are mostly unable to synthesize cholesterol and, therefore, greatly depend on glial supply, taking up these complex molecules by receptor-mediated endocytosis [20]. Additionally, oligodendrocytes, while being able to form cholesterol, preferentially use astrocyte-supplied cholesterol linked to apolipoproteins for their lipid biosynthesis [21]. 

Different mechanisms regulate cholesterol homeostasis and exchange among neural cells, which have mostly been clarified; additionally, locally released substances can interfere with cholesterol turnover and cell-to-cell transfer, as reported below. Among them, a role could be played by purines, which perform important biological functions at an intracellular level and act extra-cellularly as signal molecules [22]. Regarding the latter aspect, it has been reported that purines can interfere with the brain mechanisms that ensure proper cholesterol supply to neural cells [23,24]. 

Thus, in this review, the main topics are: (i) principal mechanisms contributing to cholesterol homeostasis in the brain; (ii) dysfunctions in these mechanisms contributing to the development of neurological/neurodegenerative diseases aforementioned; (iii) influence of purines on the cerebral cholesterol turnover and modification of purinergic signals consequent to cholesterol shuttle alterations.

## 2. Crosstalk between Neural Cells Regulates Cholesterol Homeostasis

Cholesterol homeostasis involves biochemical reactions, leading to cholesterol formation and complexation with apolipoproteins as well as an intense crosstalk between neural cells, which show the different abilities of these cells in the synthesis and request of this compound. 

### 2.1. Outline of the Brain Cholesterol Turnover

Starting from basic notions, cholesterol in the brain is predominantly synthesized in astrocytes starting from acetyl coenzyme A (acetyl-CoA), that undergo to sequential enzymatic reactions, common with peripheral tissues, ending to cholesterol formation [25]. Cholesterol is then included in high-density lipoprotein (HDL)-like particles by its major brain carrier, the Apolipoprotein E (ApoE), also synthesized by astrocytes [26]. Although the mechanisms involved in the formation of nascent lipoproteins in the brain are not fully clarified [27], the cholesterol-HDL assembly and the neo-lipid particles outside the astrocytes are regulated by members of the ABC family of transporters, such as the ATP-binding cassette subfamilies A member 1 (ABCA1) and G member 4 (ABCG4) [28].

Once secreted into extracellular space, cholesterol can be accepted by the surrounding cells, in particular neurons and oligodendrocytes. Both cell types, during brain development, have the ability of autonomously forming cholesterol that is lost when they reach maturity, even though this is not a valid for all neuronal types [29,30]. The presence of ApoE is fundamental for the neuronal uptake of cholesterol by receptor-mediated endocytosis as it behaves as a ligand of some members of the low-density lipoprotein receptor (LDLR) family (LDL receptor related protein 1 (LRP1) and LDLR) [31]. Of note, it has recently been reported that ApoE also transports miRNAs, which inhibit the expression of genes promoting cholesterol synthesis in neurons [32]. 

The amount of cholesterol in neurons must be maintained at a low level in order to not cause damage; thus, neurons provide to the intracellular metabolism of cholesterol, converting it to 24S-hydroxycholesterol (24-OHC) by the activity of the enzyme CYP46A1, also known as 24-hydroxylase [33]. The 24-OHC is then actively secreted by neurons and is taken up by astrocytes, where it activates nuclear liver X receptor (LXR) and promotes transcription of ABCA1 and ApoE [12,34]. Alternatively, 24-OHC can cross the BBB, being then cleared by the liver [35]. 

Mature oligodendrocytes also are dependent on the astroglia supply of cholesterol, differently from their precursors, which show a greater autonomy as for cholesterol biosynthesis. In these cells, the biochemical processes leading to cholesterol formation are under the control of the mechanistic target of rapamycin (mTOR) pathway, as recently reported [29]. In contrast, microglia, which are the resident brain macrophages, are more deputed to store and process sphingolipids in the physiological condition, at least [36]. However, when these cells assume a phagocytic phenotype—either during brain development to eliminate discarded cells, or in inflammatory conditions occurring also in many neurological disorders—they show upregulation of genes linked to lipid uptake and metabolism, such as lipoprotein lipases (LPLs) and ApoE [37,38]. 

### 2.2. Brain Cholesterol Transporters 

The ABC transporters are grouped in a superfamily of membrane proteins comprising seven subfamilies from ABCA to ABCG, which are deputed to the transport of various substrates across cellular membranes. 

Some members of these transporters, mainly the A subtype of the ABC proteins, have gained a great attention for cholesterol transport and implication in human diseases (reviewed in [39]). In particular, the ABCA1 protein is considered a key molecule for brain cholesterol homeostasis, as it not only exports the excess of cholesterol to lipid-free apolipropteins to generate HDL, but it also flops cholesterol from the inner to the outer leaflet of the plasma membrane. This activity, known as floppase, generates an asymmetric distribution of membrane cholesterol, which is fundamental for regulating the recruitment of intracellular signaling molecules to the membrane, and is distinct from that related to cholesterol transport. Indeed, the structure/activity relationship of ABCA1 showed that the cholesterol flop is linked to three periodically repeated leucine residues, while the VFVNFA motif is essential for both cholesterol flop and efflux [40]. Furthermore, up-regulation and silencing of the ABCA1 gene showed that this protein is involved in cholesterol efflux from astrocytes but not from neurons, similar to another ABC member, ABCG1 [41]. On the contrary, the subtype ABCG4, which is almost exclusively expressed in the brain, mainly at a neuronal level rather than in astrocytes, regulates preferentially the neuronal cholesterol efflux [19]. By the findings reported above, it is not surprising that in the CNS, alterations in cholesterol transport are associated with neuronal structural and functional deficits. For instance, alterations in the gene expression for ABCA1 transporters have recently been identified as risk factors for AD [39]. 

Other ABC proteins could be involved in the cholesterol shuttle. One of these is the less known protein, ABCA7, which is highly expressed in the CNS. However, its physiological function and, particularly, transport substrates, are today not completely elucidated. It seems that while ABCA1 and ABCA7 are equally able to export choline phospholipids, cholesterol efflux from ABCA7 was marginal [42]. This evidence indicates that the different subtypes of ABC transporters have specific preferences for molecules to be exported from neural cells. Interestingly, other genetic and functional studies have suggested a possible involvement of other two ABCA subclass members, ABCA2 and ABCA5 proteins, even though additional studies are necessary to confirm their pathophysiological role in the brain [39].

### 2.3. LDL Receptors 

Similar to their role in the periphery, LDL receptors are mediators of cholesterol exchange in the CNS. However, LDL receptors play also other roles in a wide range of cellular signaling pathways that transcend lipid metabolism, such as the modulation of synaptic function and plasticity, as well as dendritic spine formation [43]. They can also control neuronal migration and formation of cortical layers [44,45,46]. 

Focusing on the presence and function of some of these receptors in neural cells, it has been found that LRP1 is expressed in neurons, and to a lesser degree, in astrocytes, microglia and vascular smooth muscle cells [47]. Beyond the activity linked to brain lipid turnover, LRP1 interact with the postsynaptic density protein PSD-95, modulating the neuronal NMDA receptor-dependent intracellular signaling [48]. Moreover, the levels of NMDA receptor 1 and Glu receptor 1 are selectively reduced in LRP1 forebrain knock-out mice and in LRP1 knockdown neurons, which are partially rescued by restoring neuronal cholesterol [49]. Accordingly, LRP1 deletion in forebrain neurons in mice leads to a global defect in brain lipid metabolism that correlates with progressive, age-dependent dendritic spine degeneration, synapse loss, neuroinflammation, memory loss, and eventual neurodegeneration. 

Differently from LRP1, LDLR is expressed more prominently in glia than in neurons. While cholesterol levels are reduced by the deletion of the *Lrp1* gene in mouse forebrain neurons, this did not occur in *Ldlr* knockout mice. Together with LRP1, LDLR represent the primary metabolic receptors for ApoE/lipoprotein in the brain and are linked to beta amyloid (Aβ) clearance, avoiding the cerebral deposition of this peptide. Accordingly, these mechanisms are impaired in AD, thus, implying that these receptors have an important role in AD pathophysiology (reviewed in [31,50]). 

Finally, the very low-density lipoprotein receptor (VLDLR) and the low-density lipoprotein receptor-related protein 8 (LRP8, also known as ApoER2) bind triglyceride-rich apoE-containing particles such as VLDL and intermediate density lipoproteins, but not LDL [51,52]. In the brain, VLDLR is expressed in glia, neuroblasts and pyramidal neurons on cell membranes outside of lipid rafts [53,54], while LRP8 is expressed in neurons throughout the CNS and traffics postsynaptic density [55,56]. Both VLDLR and ApoER2 knockout animals show deficits in cerebellar morphology, altered contextual fear conditioning and long-term potentiation [44,57]. 

## 3. LDL Receptor Interactors in Normal and Pathological Brain Conditions 

The identification of molecules, defined as “interactors”, which interfere with biochemical processes and functional events linked to receptor activities, has become a major goal in cell biology. These interactions regulate the receptor activity/function and have high relevance for signal transduction in biological systems [58]. In this regard, multiple and bidirectional protein–protein and/or protein–carbohydrate interactions, mainly but not only involving their cytoplasmic domain, have been found to involve the receptors belonging to the LDL receptor family treated above [44,59]. The role of these interactors, which act by multiple molecular mechanisms, is outlined below. 

### 3.1. Reelin and F-Spondin 

Multiple processes involved in neuronal migration, including the early stage of radial migration and termination of migration beneath the marginal zone in the developing neocortex, are triggered by Reelin, a large secreted extracellular matrix protein through LRP8/ApoER2 receptors [60], and via VLDLR [61]. Accordingly, mice lacking functional Reelin, LRP8 or Disabled 1 (Dab1), a cytoplasmic adaptor protein adaptor implicated in neuronal development [62], show cortical, hippocampal and cerebellar neurons failing to migrate properly during brain development [63]. A similar phenotype is observed in amyloid precursor protein (APP) knockout (KO) animal models [64,65].

The binding of Reelin with LRP or VLDLR, inducing their clustering and phosphorylation of Dab1, also increases long-term potentiation in hippocampal slices [66]. Accordingly, both receptors seem to be necessary for Reelin-dependent enhancement of synaptic transmission in the hippocampus [57]. In particular, the Reelin-LRP8 pathway is required for hippocampal-dependent associative learning and is involved in the epigenomic modifications required for memory formation. In fact, LRP8-KO mice show a severe impairment in freezing behaviors that reflect a loss of long-term memory formation [67]. 

Interestingly, Reelin modulates LRP8-processing mediated by gamma-secretase, a membrane-embedded protease complex deeply involved in AD pathogenesis [68], leading to the release of the LRP8 intracellular domain (ICD) acting as a synapse-to-nucleus communication. However, LRP8-ICD is also a negative regulator of Reelin, thus, suggesting a feedback regulatory system triggered by gamma-secretase cleavage of LRP8 [69]. Apparently, this activity needed for learning and memory function is negatively regulated by ApoE4, the most important genetic risk factor for AD [48,70,71]. In contrast, Reelin signaling seems to be intact in primary neuronal cultures obtained from mice with null expression of the gene for presenilin-1 (PSEN-1^-/-^), which is the catalytic subunit of gamma-secretase [72]. As well, PSEN-1 is not required for Reelin-induced phosphorylation of Dab1 [52,73,74]. 

LRP8 is a receptor also for F-spondin, a component of the extracellular matrix involved in neuronal migration and plasticity in developmental and adult brain. This binding is able to affect APP-processing, resulting in a decreased production of Aβ peptide [75,76].

Of note, LDL receptors can additionally control the activity of glutamate receptors by modulating post-synaptic density that regulates long-term potentiation, memory and learning [77]. PSD-95, an adaptor protein directly involved in the post-synaptic density, interacts with LRP8 and NMDA receptors [68,78]. The latter are also conditioned by Reelin and LDL receptors; which, by recruiting Dab1, in turn activating Src tyrosine kinases—the phosphorylate NMDA receptor—facilitates Ca^2+^ entry through channels [79]. Interestingly, expressing a mutant of LRP8 (LRP8-tailless) altered the Reelin/LRP8/PSD-95 signaling pathway, inhibiting the interaction with PSD-95, hindering Reelin signaling and reactivating a robust dendritogenesis in mature hippocampal neurons in vitro [80].

### 3.2. APP

APP is the most known partner of ApoE receptors, and its mutations cause AD [81]. APP/ApoE receptors form macromolecular complexes that bind to extracellular matrix proteins, such as F-spondin and Reelin or metallo-proteinases of the extracellular matrix, the cleavage of which are dependent on alpha-secretase activity [53,82]. This molecular machinery seems to be positively involved in the: (i) formation of neural connection and migration during brain development; (ii) regulation of dendritic spine morphology [52,83] and (iii) regulation of the surface mobility of NR2B sub-unit of NMDA receptors [84].

The APP–LRP8 interaction seems to be also involved in the increase in cell surface APP levels, as this requires the presence of LRP8 cytoplasmic domain, resulting in a decreased APP internalization rate. Interestingly, LRP8 expression correlates with a significant increase in Aβ production and reduced levels of APP-C-terminal fragments (CTFs). The increased Aβ production is dependent on the integrity of the NPXY endocytosis motif of LRP8: expression of LRP8 increases APP association with lipid rafts and gamma-secretase activity, both of which might contribute to increase Aβ production [85].

The crosstalk between LDL receptors and APP is even more complex if we look at another interactor known as FE65, a cytoplasmic adaptor protein [86]. In fact, LRP1, LRP8 and VLDLR bind to FE65, forming different tripartite complexes, which may modulate APP endocytic trafficking, Aβ production, APP intracellular domain (AICD) nuclear trafficking and DNA protection [85,87]. Notably, AICD, along with FE65 and Tip60/Kat5, a lysine acetyltransferase regulating hippocampal gene network linked to memory formation [88], associates into complexes in nuclear transcription factories, while the C-terminal portions of LDL receptors are recruited at a nuclear level upon gamma-secretase cleavage [67,89]. At the same time, AICD, along with FE65 and Tip60/Kat5, blocks LRP1 transcription [90], suggesting that APP proteolytic processing by gamma-secretase reduces LRP1 brain levels, possibly hampering both Aβ clearance and Reelin signaling.

### 3.3. Syntaxin 5 (Stx5), Selenoprotein P, Beclin 1 and NYGGF4

Several other extracellular and intracellular interactors of apolipoprotein receptors modify the processing of the LDL receptor proteins, thus, changing their functions and trafficking either to or from the cell surface.

For instance, syntaxin 5 (Stx5) is an interactor protein partner for some LDL-receptor sub-types involved in intracellular vesicle trafficking [91].

Selenoprotein P, secreted from the liver as an exosomal component to be protected from plasma proteases and transported from hepatocytes to neuronal cells, interacts with ApoE through its heparin-binding sites and co-localizes with the ApoER2, thus, contributing to regulate the exosomal secretion [92].

Instead, Beclin 1 is a component of a group of agents with stimulating or suppressive activities, representing an interactome system which regulates the initiation of the autophagosome formation. The expression of Beclin 1 is reduced in AD patients; its deficiency or caspase-dependent enhanced proteolytic cleavage favors the deposition of Aβ, whereas its overexpression reduces the accumulation of Aβ peptides [93].

Moreover, several other cytoplasmic proteins act as positive interactors of LRP receptors, in particular of the LRP1 subtype, which principally modulates the formation and vascular clearance of Aβ [94,95]. They include:-Jnk interacting protein-1B (JIP1B), a scaffold protein largely present in the brain which assures a fast and efficient APP anterograde transport in neurons [96];-end-binding 1 (EB-1) protein 1, an important controller of microtubule dynamics [97];-NYGGF4, also known as phosphotyrosine interaction domain containing 1 (PID1), which shows a specific binding with LRP1 and whose expression is decreased in AD patients [98,99].

## 4. Influence of the Purinergic Signaling on the Main Functions of the Cholesterol Shuttle

Purines are ancestral molecules, ubiquitously present inside the cells, where they contribute to vital functions such as cell duplication, protein formation and energy supply. Furthermore, these compounds are present in the extracellular fluids, where they interact with own receptors [100], thus, activating the so-called “purinergic signaling”. This is a ubiquitous system of cell-to-cell communication, whose final effects depend on the type and number of natural ligands, which are released from cells or extracellularly formed by metabolizing ecto-enzymes (see Figure 1 in which the release of adenine-based purine release and their extracellular metabolism are outlined).

The purines, which are usually detectable in the extracellular fluid of virtually all cell types and can act as natural ligands of specific receptors, comprise adenine- and guanine-based compounds, such as adenosine or guanosine tri-, di- and mono-phosphate (i.e., ATP, ADP, AMP, GTP, GDP, GMP) as well as their derivatives (adenosine and guanosine) and some uridine-based nucleotides (UTP, UDP). 

Different purine receptors have been so far cloned and detected on cell membranes, which are mostly grouped into two big families. The P1 receptor (P1R) family comprise four metabotropic subtypes, namely A_1_, A_2A_, A_2B_ and A_3_ receptors, which respond to adenosine and are coupled to different G proteins and downstream molecular pathways. The P2 receptor (P2R) family is divided into two further subgroups; in particular, seven P2X ligand-gated ion channels which are responsive to ATP and belong to the first subgroup, while eight P2Y metabotropic receptors, which are activated mainly by ATP and some of them also by ADP or UTP/UDP, belong to the second one (see Table 1 and related legend). In addition, currently, there is not a completely characterized adenine-selective P0 receptor, while specific receptor sites for guanine-based purines still need to be identified [103,104].

P1 receptors are linked to multiple types of G proteins. In particular, A_1_R and A_3_R are coupled to both G_i/o_ proteins, the activation of which leads to the inhibition of adenylate cyclase (AC) activity and cAMP formation, and G_q_ proteins, which in turn activate phospholipase C (PLC), thus, increasing inositol triphosphate (IP3) and diacylglycerol (DAG) formation with an intracellular calcium level increase and protein kinase C (PKC) activation, respectively. On the other side, A_2A_R and A_2B_R are coupled to G_s_ proteins, the activation of which leads to an increase in AC activity and cAMP formation. A_2B_R also shows the ability to activate G_q_ proteins, with the consequent increase in PLC activity, as mentioned above. Moreover, all P1 receptors stimulate mitogen-activated protein kinase (MAPK) pathways, including extracellular signal regulated kinase 1 (ERK1), ERK2, Jun N-terminal kinase (JNK), and p38-MAPK. Of note, A_1_R and A_2_AR exhibit the greatest affinity towards ADO, whereas A_2B_R and A_3_R exhibit the lowest [103,105]. The P2 receptor (P2R) family is subdivided into seven ionotropic P2X (P2XR), which are activated by ATP, and eight metabotropic P2Y receptors (P2YR), of which P2Y_1_R respond to ATP and ADP; P2Y_2,4,6_R are mainly activated by uridine-based nucleotides, whereas P2Y_12,13_R respond to ADP and P2Y_14_R to UDP-glucose [106,107]. P2Y_1,2,4,6_R are coupled to Gq proteins and PLC, the activation of which causes the downstream effects mentioned above for P1 receptors. In contrast, P2Y_12,13,14_R activate G_i_ proteins, leading to the inhibition of AC and reduction in cAMP levels, whereas P2Y_11_R stimulation causes an increase in intracellular Ca^2+^ and cAMP levels by activating both G_q_ and G_s_ proteins. Finally, P2YR can also recruit the βγ subunit of G proteins, with activation of multiple effectors, such as phosphatidylinositol-4,5-bisphosphate 3-kinase γ (PI3K-γ), phospholipase C-β2 and -β3, inward rectifying K^+^ (GIRK) channels, G protein-coupled receptor (GPCR) kinases 2 and 3, Rho, and MAPKs. On the other side, ionotropic P2XR are formed by three subunits assembled to form homo- or heterotrimers and, when stimulated, allow the entry of cations such as Na^+^ and or Ca^2+^ into cells. Among them, P2X_7_R show the lowest affinity towards ATP (around 100 µM) and their prolonged stimulation by the ligand opening a macropore, allowing the entry of large molecules up to 900 Da [103].

So far, the interaction between purinergic signals and the cholesterol shuttle has been poorly investigated. The main findings about this aspect are reported below.

### 4.1. Metabotropic P2 Receptors

Starting from P2Y_1_R, they are expressed in the brain, playing a major role in the activities of astrocytes [108]. In chronic neurodegenerative conditions such as AD, astrocyte tasks are profoundly altered, showing electrical hyperactivity with an increased frequency of spontaneous calcium events [109], which seem to be mediated by the hyperexpression of P2Y_1_R and downstream effectors. Accordingly, it has been reported that the chronic inhibition of these receptors reduced neuronal-astrocyte network hyperactivity, also restoring hippocampal synaptic integrity and improving learning and memory in AD mice. However, the beneficial effects consequent to P2Y_1_R inhibition were not related to changes in Aβ deposition mechanisms, but rather to a modulation of the astrocytic barrier capable of restraining the Aβ toxic effects on neurites surrounding the amyloid plaques [110]. Thus, P2Y_1_R are not directly involved in cholesterol shuttle regulation, although some papers seem to support a possible role. Indeed, it has recently been reported that immortalized *APOE4* versus *APOE3* astrocytes show increased Ca^2+^ excitability, together with an altered membrane lipidome pattern and intracellular cholesterol distribution. This astrocyte hyperactivity, however, was exhibited by astrocytes from male, but not female, mice with ApoE targeted replacement [111]. The study suggests that, independent of Aβ-induced brain alterations, the abnormal astrocyte excitability, likely sustained by P2Y_1_R activity, is also related to altered cholesterol homeostasis. Clearly, further investigation needs to confirm these findings and to clarify the dependence of the above events on sexually-related differences. 

Additionally, P2Y_1_R activity might be involved, together with other P2YR expressed at the CNS, in a process similar to the “reverse cholesterol transport” (RCT) observed in peripheral tissues. This process concerning the cholesterol efflux from peripheral cells as well as its transport to the liver for further metabolism and biliary excretion is mediated by HDL, thus, supporting the anti-atherogenic effect of these lipoproteins [112]. Among the cellular partners necessary to assure the HDL functions, there is the ecto-F1-ATPase, whose activity is coupled to some P2Y receptor subtypes which are crucial molecular components in this pathway [113]. In more detail, the stimulation of ecto-F1-ATPase in the liver by apoA-I triggers a low-affinity-receptor-dependent HDL endocytosis by a mechanism strictly related to ADP generation. Accordingly, the concomitant stimulation of P2Y_1_R or P2Y_13_R, the main P2Y receptors present in hepatic cultured cells, promotes HDL endocytosis [114,115,116]. As well, on endothelial cells, ecto-F1-ATPase is activated by HDL-apoA-I complex and is potentially coupled to the activation of P2Y_1_R or P2Y_12_R, which promotes cell survival and HDL transcytosis [117]. Looking at the CSN, it is known that astroglial cells, beyond the free cholesterol (FC), are able to secrete lipoprotein particles which are donated to other neural cells with a process that could be called “Brain Reverse Cholesterol Transport—BRCT”, as proposed by Yu and coll. [118]. Thus, it would be of interest to investigate whether P2Y_1_R, with the help of other P2YR (i.e., P2Y_13_R), may act in the brain similar to in the periphery. This research could also be stimulated by the evidence that a deficiency of palmitoyl protein thio-esterase 1 (Ppt1), which causes an infantile form of neuronal ceroid lipofuscinosis (INCL), is associated in mice with increased ecto-F1-ATPase expression and apoA-I uptake in neurons, while serum cholesterol and apoA-I concentrations were decreased. In addition, in sera from Ppt1-deficient mice apoE-containing HDL, particles were almost totally absent, thus, indicating functional changes in the RCT pathway [119]. 

The activity of other purine receptors belonging to the P2YR family affect the brain cholesterol homeostasis. For example, in human 1321N1 astrocytoma cells, the activation of P2Y_2_R enhanced the release of sAPP alpha, the non-amyloidogenic product of APP with neuroprotective properties, deriving from the alpha-secretase activity. This effect involved the activity of some metalloproteases but not a trans-activation of epidermal growth factor receptor (EGFR) [120]; instead, it happens in smooth muscle cells from artery segments. In these cells, indeed, the stimulation of the P2Y_2_R is through an EGFR-transregulated mechanism coupled to RhoA and Rac1 GTPase activation. Of note, this mechanism quickly affected the plasma membrane redistribution of P2Y_2_R by displacing it within membrane rafts and favoring its rapid internalization [121].

Similar to in astrocytoma cells, in rat primary cortical neurons (rPCNs), the IL-1β-induced up-regulation of P2Y_2_R, virtually absent in quiescent cells, enhanced the production of sAPP alpha, suggesting that pro-inflammatory stimuli can enhance non-amyloidogenic APP-processing through the P2Y_2_ sites also in neurons [122]. The G_q/11_-PLC-Ca^2+^ signaling cascade activated by the P2Y_2_R stimulation seems to be crucial in the membrane lipid raft mobilization and in the mouse neuroblastoma x rat glioma hybrid cell line migration [123].

### 4.2. Ionotropic P2 Receptors and Brain Cholesterol Turnover

Seven subtypes of P2X receptors (P2X_1-7_R) have been identified, which are ubiquitously expressed in mammals. All of them, except P2X_6_R, are assembled as functional homotrimers [124]. P2X_1_R and P2X_3_R show the highest apparent affinity to ATP, which activates them at concentrations below 1 µM, while P2X_7_R display the lowest affinity towards the nucleotide (over 100 µM). The receptor kinetics are also different. Indeed, activation of P2X_1-4_R is faster than that of P2X_5_R and P2X_7_R; while desensitization of P2X_1_R and P2X_3_R is fast, that of P2X_2_R and P2X_4_R is moderate or slow and P2X_7_R does not desensitize. It should also be highlighted that while all P2XR are non-selective cation channels that allow the entry of monovalent ions such as Na^+^, K^+^, and Li^+^, Rb^+^ or Cs^+^, and divalent ions such as Ca^2+^, P2X7Rs P2X_7_Rs, when exposed to a sustained ligand stimulation, widen their channels forming a “megapore” permeable to organic molecules much larger than ions [124]. 

Notably, most P2XRs are stably expressed at the plasma membranes and their trafficking and function are regulated by membrane proteins and lipids, with which they interact. In particular, several P2XR are associated with lipid rafts, which are membrane microdomains enriched in lipids such as cholesterol, sphingolipids and saturated phospholipids, whose levels can be variable among cells. Thus, regulation of lipid rafts provides a mechanism for changing the functional expression of P2XR [125]. For instance, plasma membrane cholesterol depletion with methyl-β-cyclodextrin alters lipid raft composition as well as the function of some (i.e., P2X_1,3,4_Rs), but not all of the P2XR. In patch clamp studies, the cholesterol depletion from lipid rafts reduced currents evoked by ATP stimulation of several P2XR subtypes (except that of P2X_2_R and P2X_4_R), without changing the ATP sensitivity or their cell surface expression. This suggests that cholesterol is normally needed to facilitate the opening/gating of ATP-bound P2X receptor channels [126]. Additionally, some P2XR, such as P2X_2_R, can be regulated in their targeting to synapses by the APP-binding protein, Fe65 [127].

In this context, a particular attention should be paid to the function of P2X_7_R. Indeed, cholesterol depletion potentiates the activity of these ATP low-affinity receptors and modifies the gating mechanisms of the pore associated to the ionic channel of this receptor [128]. Conversely, the stimulation of P2X_7_R regulates the lipid raft composition by promoting the activity of lipid-metabolizing enzymes, including phospholipases and sphingomyelinases [125]. 

However, the roles played by P2X_7_R are not always univocal in several brain cholesterol-related mechanisms and functions [129,130]. Among these, it should be mentioned that P2X_7_R activation has been implicated in neuroinflammation coupled to neuronal damage and death. Indeed, sustained activation of P2X_7_R stimulates the production of reactive oxygen species (ROS) that trigger Aβ peptide formation, thus, contributing to cause AD-like neuronal damages [131]. At the same time, Aβ_1-42_ peptide, one of the most known AD biomarkers, induces microglial cells to release ATP that produces ROS via P2X_7_R activation [132], thus, creating a vicious circle [106]. Moreover, it has been reported that APP-processing is influenced by the ATP-mediated activity of P2X_7_R, able to counteract AD pathogenesis. Indeed, although in AD patients, the main product of APP proteolytic process is Aβ, which derives from beta- and gamma-secretase activity, APP can be alternatively cleaved by alpha-secretase, leading to the formation of nonpathogenic Aα (Aα) peptides [133]. Regarding these events, it was found that a reduced P2X_7_R expression or the administration of P2X_7_R antagonists increased alpha-secretase activity through the inhibition of glycogen synthase kinase-3β (GSK-3β) and decreased the number of amyloid plaques [134,135,136]. However, single nucleotide polymorphisms of these receptors have been reported in neurological disorders including AD (reviewed in [137]). Thus, P2X_7_R effects seem to depend on different factors, also including the amount of available APP and multi-factorial-dependent unbalance between alpha- and beta/gamma-secretases.

### 4.3. Metabotropic P1 Receptors and Brain Cholesterol Turnover

In the CNS, adenosine is recognized as an important neuromodulator, being involved in the control of the synaptic plasticity, neuronal survival and fundamental cerebral activities such as motor function, cognition and sleep [138]. As aforementioned, adenosine can activate different metabotropic receptors, among which the most expressed in the brain are the A_1_R and A_2A_R. 

Of these, the main function of A_1_R is neuromodulation, as their activation induces sedation, diminishes anxiety, inhibits seizures and reduces neuronal damage consequent to different brain injuries [139]. However, apart from a decrease in the number of A_1_R in the dentate gyrus of AD patients, a role in cholesterol control has not been reported for these receptors [140]. 

In contrast, the A_2A_ receptors (A_2A_R) seem to have particular relevance in the interactions between astrocytes and myelinated neurons [141]. Indeed, a number of results have demonstrated that the A_2A_R activity modulates the differentiation and migration of oligodendrocytes, which are the myelinating cells in the CNS. In particular, in the neurodegenerative NPC disease, in which there is an accumulation of cholesterol and glycolipids in the lysosomes of oligodendrocytes [142], in parallel with a large dysfunction of the operation of these cells, a reduced level of adenosine has been shown in the brain of NPC1^-/-^ mice; this was an experimental model of the disease, which was in turn coupled to impaired synaptic plasticity and cognitive deficits of the same animals [143]. It was also demonstrated that in fibroblasts from NPC1 patients, the A_2A_R stimulation by the agonist CGS21680 restored the cholesterol distribution inside the cells together with lysosomal calcium content and mitochondrial membrane potential [144]. Further experiments in human neuronal and oligodendroglial cell lines, in which the NPC1 phenotype was induced by small-interference RNA, the A_2A_R stimulation reduced the harmful cholesterol accumulation, while normalizing mitochondrial membrane potential [145]. Accordingly, in NPC1^-/-^ mice, an increase in the brain adenosine level obtained by inhibiting its transport inside the cells significantly improved the animal cognitive deficits while reducing Purkinje neuron loss and sphingomyelin accumulation in the liver [146]. As for oligodendrocyte progenitor maturation, the exposure of primary cultures of these cells to U18666a, an inhibitor of cholesterol transport used to induce NPC-like phenotypes in vitro, caused an intracellular accumulation of cholesterol, abnormal mitochondrial depolarization and impaired autophagy. Interestingly, treatment with CGS21680 counteracted the arrest in cell maturation arrest induced by U18666a and restored the complex cell morphology [147]. In addition, a reduction in membrane cholesterol levels occurring in NPC1 hinders the normal A_2A_R activity with a decrease in cyclic adenosine monophosphate (cAMP) production [148]. Altogether, these findings indicate that there is an important crosstalk between A_2A_R and the cholesterol shuttle, in that the A_2A_R activity can be impaired by intracellular cholesterol accumulation, as observed in NPC1 cells, while the A_2A_R stimulation may hinder cholesterol accumulation and mitochondrial dysfunction, promoting oligodendrocyte differentiation. These data suggest that A_2A_R modulation could represent a promising therapeutic tool for NPC disease. 

Additionally, in AD, it was shown a protective role of A_2A_R which colocalize with gamma-secretase complexes in endosomes and interact with the catalytic subunit of this enzyme that is PS1. Interestingly, istradefylline, the antagonist for these receptors, increased Aβ generation in primary neuronal cells from the AD mouse model. Likewise, the knockdown of A_2A_R potentiated both Aβ generation and gamma-secretase activity, suggesting a positive modulatory role of A_2A_R activity on gamma-secretase activity [149].

Interestingly, a recent review indicates the use of radioligands for A_2A_R as a tool for PET imaging for the study of the in vivo expression of these receptors in neuroinflammatory and neurodegenerative diseases, including AD [150,151].

### 4.4. Guanosine and Cholesterol Turnover

Guanosine (GUO) belongs to the family of the guanine-based purines (GBPs), which comprises other pharmacologically active molecules such as guanosine triphosphate and monophosphate (GTP and GMP, respectively), and the GUO metabolite, guanine (GUA). Numerous studies have shown that GBPs are neuroprotective agents contributing to nervous tissue repair upon brain injury (reviewed in [105,152,153]). GUO, in particular, exhibits anticonvulsant, antiparkinsonian, anxiolytic and antidepressant effects [152,153]. Additionally, some findings from our group have shown that GUO can stimulate the efflux of cholesterol and the expression of the ApoE in rat brain-cultured astrocytes and in C6 rat glioma cells by the involvement of the PI3K/ERK1/2 pathway [154]. These results corroborate the neuroprotective activity of GUO at the CNS; indeed, GUO, by increasing the availability of the glial-derived cholesterol/ApoE amount at an extracellular level might contribute to assure the normal synaptic activity. Furthermore, the GUO stimulation of rat-cultured astrocytes increased both the release of UTP and the expression of P2Y_2_R, which can exert a protective role based on the findings reported above about astrocyte UTP/P2Y_2_R signaling against AD risk [155].

## 5. Discussion

Defects in brain cholesterol metabolism has been shown to be implicated in neurodegenerative diseases, such as Alzheimer’s disease (AD), Huntington’s disease (HD), Parkinson’s disease (PD), Niemann–Pick type C disease and some cognitive deficits typical of the old age [16,156,157]. LDL receptors are ubiquitous key players in cholesterol homeostasis; however, there is a large area of knowledge that has not yet been adequately explored, especially the one concerning the involvement of these receptors with neuronal or astrocytic localization in the turnover of cerebral cholesterol. 

This molecule, as reported above, has unique metabolic aspects, and is regulated in a manner largely independent from the peripheral systems, making the CNS a completely isolated enclave regarding the regulation of membrane lipids and receptor systems concentrated in lipid rafts. As reported in the previous chapters, there is a plethora of molecules interacting with cholesterol homeostasis and LDLR functions. Dysregulation in the activity of these interactors has important implications on the development of brain pathologies, of which one of the most studied is the AD amyloidosis, a neurodegenerative event restricted to the brain, although the genes and proteins responsible for amyloidosis (APP, PS) are ubiquitous, even in familial cases. 

So far, poor attention has been paid to the involvement of purinergic receptors and their signaling in the metabolism of cerebral cholesterol and its complex homeostasis. In addition, the effect that LDL receptors and their turnover exert on membrane lipid compositions and, in turn, on the physiological and pathological regulation of purinergic receptors, is very limitedly explored. Here, trying to put together the pieces of a puzzle, a picture has emerged in which some purinergic receptors, directly or indirectly, interact with the LDL receptor systems, as summarized in Table 2. 

Although the precise mechanisms of these interactions as well as the related pathological implications are currently poorly understood, the purine receptors seem to be involved in the pathogenesis of neurodegenerative diseases. In particular, in AD, the most evident signs concern both the involvement of some purine receptors (A_2A_R, P2X_7_R and P2Y_2_R) in the endocytosis and processing of LDL receptors, and APP in the cholesterol turnover and in the Aβ peptide generation. Moreover, guanosine, albeit in lack of a specific receptor, seems to assure a normal neuronal activity by enhancing cholesterol and ApoE efflux from astrocytes. Finally, it is worth mentioning the important crosstalk between the A_2A_R and cholesterol shuttle in oligodendrocytes, which can be altered in NPC disease, the neurodegenerative disorder mainly affecting cholesterol turnover in these cells. 

In summary, LDL receptors and purinergic receptors hide an interesting potential in the pathophysiology of the CNS. Their role in the brain cholesterol lifetime as well as their interaction with protein complexes, whose enzymatic/proteolytic activity leads to the formation of peptide fragments of different sizes and specific sequences, requires to be elucidated and framed within the appropriate context. In particular, in a scenario where LDL receptor roles are clear in the cholesterol metabolism and in the amyloidosis, given their interaction with APP, the corresponding roles of the purinergic receptors need to be further clarified. These answers would help to define a bigger “picture” of the complex functional mechanisms involving the partners of the “cholesterol shuttle” in neurodegeneration.

## Figures and Tables

**Figure 1 ijms-23-08683-f001:**
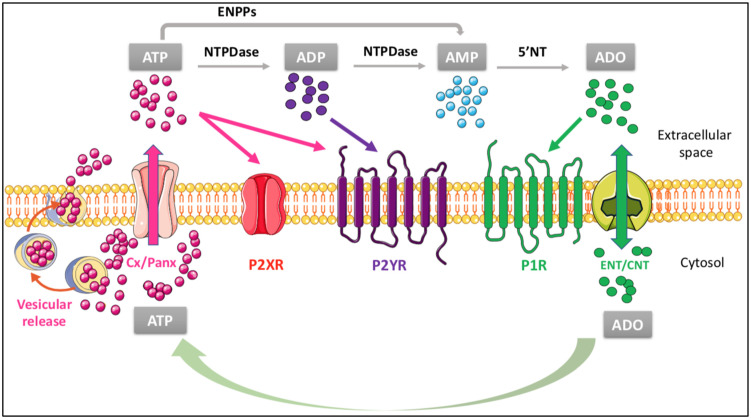
In normal conditions, ATP is released from virtually all cells by multiple ways, i.e., vesicular exocytosis, facilitated diffusion by nucleotide-specific ATP-binding cassette (ABC) transporters, connexin/pannexin (Cx/Panx) hemichannels and multiple organic anion transporters. In contrast, adenosine is mainly generated from the extracellular nucleotide metabolism while adenosine efflux from cells usually occurs under cell stress conditions by selective carriers (ENT/CNT) [101]. Purines are also extracellularly metabolized. The principal family of ATP-metabolizing enzymes are the nucleoside triphosphate diphosphohydrolases (NTPDases), which consist of eight members, of which NTPDase 1, 2, 3, and 8 are cell surface-bound enzymes, with different activities. Extracellular ATP can also be metabolized by enzymes belonging to the family of ectonucleotide pyrophosphatases (ENPPs) and acid phosphatases (APs). AMP, derived from ATP metabolism, is degraded to adenosine (ADO) mainly by ecto-5′-nucleotidases (5′-NT, also known as CD73). Once formed, ADO can be further metabolized at an extracellular level up to hypoxanthine by the combined activity of cell surface-located enzymes, i.e., adenosine deaminase (ADA) or purine nucleoside phosphorylase (PNP), or transported into the cell by specific transporters to fill the intracellular pool of adenine nucleotides [102]. ENT: equilibrative nucleoside transporter; CNT: concentrative nucleoside transporter.

**Table 1 ijms-23-08683-t001:** Main characteristics of the most known purinergic receptors.

Purine Receptors			
	P1 Receptors	P2 Receptors	
Receptor subtypes	Metabotropic receptors	Metabotropic P2Y receptors including	Ionotropic P2X receptors including
	A_1_, A_2A_, A_2B_, A_3_	P2Y_1_, P2Y_2_, P2Y_4_, P2Y_6_, P2Y_11_, P2Y_12_, P2Y_14_, P2Y_14_	P2X1, P2X2, P2X3, P2X4, P2X5, P2X6, P2X7
		P2Y_1_, P2Y_2_, P2Y_4_, P2Y_6_, P2Y_11_, P2Y_12_, P2Y_14_, P2Y_14_	
Ligand(s)	Adenosine (ADO)	ATP, ADP, UTP, UDP	ATP
Downstream effectors	Coupling to different types of G proteins and molecular pathways	Coupling to different types of G proteins and molecular pathways	Ion channels whose activation allows cation entry

**Table 2 ijms-23-08683-t002:** Reciprocal interactions between purinergic signals and cholesterol turnover.

(A) *Activities* *of purinergic receptors in relation to the brain “cholesterol shuttle”*			
Purine Signal	Cell Type	Activity on Cholesterol Shuttle and Related Dysregulation	Ref.
P2Y_1_R	Immortalized *APOE4* astrocytes	Increased Ca^2+^ excitability coupled to an altered lipidome pattern and intracellular cholesterol accumulation, possibly related to receptor hyperactivity.	[111]
P2Y_1_R in cooperation with other P2YR (P2Y_12_R or P2Y_13_R)	Brain neural cells	To be explored in relation to a possible increase in the activity of neuronal ecto-F1-ATPase and ApoA-I uptake, similar to that observed in the liver or endothelial cells to assure the process known as “brain reverse cholesterol transport” (BRCT).	[118,119]
P2Y_2_R	Human 1321N1 astrocytoma cells	Increased release of sAPPα deriving from the alpha- secretase activity. It should be investigated if this effect may provoke P2Y_2_R redistribution/internalization, as observed in peripheral cells.	[120,121]
	Rat cortical neurons upon IL1β stimulus	Receptor up-regulation coupled to an increased production of the protective sAPP alpha.	[122]
	Rat cultured astrocytes	Receptor expression increased by cell stimulation with GUO, which also enhanced the UTP release from these cells, thus, contributing to the protective activity of astrocyte UTP/P2Y_2_R against AD risk.	[155]
P2X2R	CA1 hippocampal pyramidal cell/Schaffer collateral synapses	Interaction of the beta-amyloid precursor protein-binding protein Fe65 with the receptor at postsynaptic excitatory synapses, which resulted in receptor activity inhibition.	[127]
P2X7R	Neural cells	Altered membrane lipid raft composition consequent to receptor stimulation.	[125]
		Increased ROS formation, which in turn, triggered Aβ peptide formation.	[131]
	Microglia	ATP release induced by Aβ_1-42_ peptide, which in turn, stimulated P2X_7_R -related ROS production.	[132]
	Neural cells	P2X_7_R inhibition increased alpha-secretase activity, diminishing the number of amyloid plaques.	[134,135,136]
	AD	Receptor polymorphisms.	[137]
A_2A_R	Fibroblasts from NPC1 patients as well as in human neuronal and oligoglial cell lines, in which NPC1 phenotype had been induced by siRNA	Receptor stimulation reduced the harmful intracellular accumulation of cholesterol as well as mitochondrial damage.	[144,145]
	Primary culture of oligodendrocyte progenitors	Receptor stimulation counteracted the cell maturation arrest induced by the inhibition of cholesterol transport and restored cell morphology.	[147]
	Primary neuronal cells from AD mouse model	Istradefylline, a receptor antagonist, increased Aβ generation as well as A_2A_R KO-potentiated Aβ generation and gamma-secretase activity	[149]
** *(B) Influence of cholesterol turnover modifications on purinergic signals* **			
**Cholesterol turnover alterations**	**Cell type**	**Modification of the purinergic signal**	**Ref.**
Membrane cholesterol depletion	Neural cells	Reduction of calcium currents induced by P2X_2,4_R stimulation, likely related to the expression of these receptors within membrane lipid rafts.	[125,126,127]
		Potentiated receptor activity caused by membrane cholesterol depletion as well as P2X_7_R stimulation.	[128]
Reduction in membrane cholesterol levels	NCP1 cells	Impairment of normal A_2A_R activity with a decrease in cyclic adenosine monophosphate (cAMP) production.	[148]

## Data Availability

Not applicable.

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
