# Peer review of "Bidirectional Control between Cholesterol Shuttle and Purine Signal at the Central Nervous System"

_ijms, 2022, doi:10.3390/ijms23158683_

Round 1
Reviewer 1 Report
The authors of this review summarize recent knowledge about mechanisms controlling the formation and turnover of cerebral cholesterol. Especially, the authors provide detailed description of different lipoprotein receptors, their interaction with specific neuronal proteins and purinergic P1 and P2 receptors, with the aim to understand the role of purinergic signaling in cholesterol homeostasis and neurodegeneration. This is a very well written and very informative review.
Minor comments:
Figure 1 and Table 1 show in principle only basic information about purinergic signaling - extracellular ATP degradation and usual classification and signaling pathways of ionotrophic and metabotrophic purinergic receptors, without any relationship to cholesterol turnover. The authors should consider to include a scheme showing “cholesterol shuttle“ with highlighted components involved in cholesterol-purine reciprocal control, which is the aim of this review.
Table 1: “Ion channels…”this text should be better arranged
Line 29: “some purinergic receptors”. Please, specify.
Line 350: „A1R", this abbreviation is not used consistently throughout the text, sometimes „1“ is a subscribe (see line 357, for example)
Line 137: „The ATP-binding cassette (ABC) …“ abbreviation was already introduced (see lines 75 and 106)
Line 353: PKC
Lines 365-366: „P2XR are formed by at least three homo- or hetero-subunits …“ P2X are trimers, as evidenced from crystal structures. This text should be rewritten maybe like this: "Three P2X subunits can assemble to form homo- or heterotrimers …"
Line 439: P2X7 does not desensitize; the P2X4 and P2X2 desensitize moderately or slowly.
Author Response
We thank this Reviewer for his/her suggestions. We revised the entire manuscript trying to fulfill the requests. In relation to the suggestion of introducing a scheme to show the cholesterol-purine reciprocal control, it was decided to introduce a new table (table 2) as the scheme was too complicated as the mechanisms involved were multiple. We hope this solution will be acceptable to this reviewer. All changes to the manuscript are highlighted in yellow for ease of review.
Reviewer 2 Report
The Review conducted by Passarella and colleagues systematically expounded on the mechanism of such metabolism in the central nervous system for some neurometabolic diseases and specifically discussed the interaction mechanism of cholesterol and purine. The author addressed this review as comprehensive and systematic references, most of the cited papers are within 5 years, so this work is very convenient for readers to read and can help readers better understand related fields. This is a qualified review.
Author Response
We thank this Reviewer for the appreciation of our paper.